# Root Herbivory: Grass Species, *Epichloë* Endophytes and Moisture Status Make a Difference

**DOI:** 10.3390/microorganisms8070997

**Published:** 2020-07-03

**Authors:** Alison J. Popay, Joanne G. Jensen, Wade J. Mace

**Affiliations:** 1AgResearch, Ruakura Research Centre, Hamilton 3240, New Zealand; joanne.jensen@agresearch.co.nz; 2AgResearch, Grasslands Research Centre, Palmerston North 4410, New Zealand; wade.mace@agresearch.co.nz

**Keywords:** *Epichloë coenophiala*, *Epichloë uncinata*, *Festuca arundinacea*, *Festuca pratensis*, *Costelytra giveni*, drought stress, loline alkaloids, symbiosis

## Abstract

The root-feeding scarab insect *Costelytra giveni* causes severe damage to pasture ecosystems in New Zealand. Loline alkaloids produced by some *Epichloë* endophytes deter this insect. In two experiments, tall fescue infected with *E. coenophiala*, strain AR584, and endophyte-free (Nil) controls were subjected to pulse drought stress (DS) or well-watered conditions (WW). The second experiment also included meadow fescue infected with *E. uncinata*. After 4–6 weeks exposure to the different conditions, roots were excised and fed to *C. giveni* larvae for 7 days. Relative root consumption (RC), frass production, and relative weight change (RWC) of larvae were measured and the loline content of roots determined. RC and frass output were higher for larvae feeding on Nil DS tall fescue than WW and reduced by AR584. RWC was also greater on DS than on WW Nil plants but reduced by endophyte only in DS plants. RC, frass output, and RWC of larvae were reduced by endophyte in DS and WW meadow fescue, but the effect was greater for WW plants. Loline alkaloid concentration in roots was significantly higher in DS than WW tall fescue in Experiment I but higher in WW than DS meadow fescue in Experiment II. These experiments have demonstrated that moisture status interacts with endophyte to differentially affect root herbivory in tall fescue and meadow fescue.

## 1. Introduction

Interactions between an insect herbivore and its host plants are complex and governed by a multiplicity of factors. The complexity increases when obligate biotrophic symbionts that influence herbivory are involved such as *Epichloë* fungal endophyte species that live systemically in above-ground tissues of grasses (Pooidae). Overlay environmental factors and a network of interactions becomes possible and indeed likely. Both tall fescue (*Festuca arundinacea* Schreb.) and meadow fescue (*Festuca pratensis* Huds.) naturally host the endophytes, *E. coenophiala* and *E. uncinata* respectively. These endophytes have effects on a range of herbivores due to the production of loline alkaloids, particularly *E. uncinata* which produces much higher levels of loline alkaloids in herbage by comparison with other loline-producing associations such as tall fescue with *E. coenophiala* [1]. The selected *E. coenophiala* strain, AR584, used in the trials reported here has a loline profile that slightly differs from that of *E. uncinata* in that *N*-acetylnorloline (NANL) is the dominant form whereas *N*-formylloline (NFL) is the major component alkaloid in meadow fescue. The AR584 genotype also produces peramine but not ergovaline, the cause of fescue toxicosis in grazing animals.

Unlike most of the known alkaloids produced by endophytes, lolines are translocated into the roots where they are known to affect root feeding insects. Evidence for effects of endophyte in tall fescue on scarab grubs have been mixed. Controlled studies have shown reduced survival and growth of neonate larvae of *Popillia japonica* and *Cyclocephala lurida* and increased tolerance to herbivory [2,3,4] due to the common toxic endophyte but other studies have shown little or no effects [2,5,6]. There is, however, evidence that *E. uncinata* in meadow fescue reduces larval growth of *C. giveni* and *Heteronychus arator* in the field and in bioassays [7,8].

Both *E. uncinata* and *E. coenophiala* are known to reduce the impact of drought on their host plants [9,10,11]. Although the mechanisms are not well understood, reduced stomatal conductance and leaf water potential have been implicated as mechanisms behind the greater tolerance but protection from oxidative stress and larger root systems may also have a role [11,12]. Positive growth responses resulting from reduced herbivory mediated by *Epichloë* endophytes have been well documented in ryegrass [13,14,15] while improved performance of tall fescue and meadow fescue in New Zealand and Australia has been attributed to endophyte mediated protection from both biotic and abiotic stress [16,17]. Enhanced plant growth facilitates access to more resources through photosynthesis and uptake of water and nutrients [12,18,19]. These factors alone provide a degree of drought resistance and/or tolerance.

The interactions between moisture status, endophyte, and insect herbivory has received relatively little attention. Bultman and Bell [20] found both *Rhopalosiphum padi* aphids and the fall armyworm (*Spodoptera frugiperda*) were affected in different ways by a combination of drought and endophyte. Aphid density was lower on endophyte-infected than on endophyte-free plants and lower on endophyte-free plants that were droughted compared to well-watered. On the other hand, fall armyworm fed harvested stems and leaf blades from drought-stressed plants exhibited reduced growth and development compared with larvae fed the same material from well-watered plants. No similar effect was shown with endophyte-free plants. Similarly, in their study of *L. multiflorum*, Miranda et al. [21] found that endophyte reduced aphid densities but only on drought stressed plants. In field studies, endophyte-mediated protection from herbivory during drought can prevent plants from reaching a tipping point from which they do not recover [13,15,22].

Here we report on an experiment in which tall fescue plants with and without *E. coenophiala* strain AR584 were watered or drought stressed in a screenhouse and a glasshouse to determine if differences in temperature also had an effect. A similar watering regime was applied in a second experiment to both tall fescue and meadow fescue infected respectively with *E. coenophiala* and *E. uncinata*, or endophyte-free. Because plant genotype is recognized as a factor in the physiological and biochemical responses of tall fescue/endophyte associations to drought [23,24,25], plant genotypes that were either endophyte-infected or endophyte-free in each trial were cloned across treatments. After 4–6 weeks of differing watering regimes, roots were harvested and fed to *Costelytra giveni*, an endemic New Zealand insect pest, in a bioassay over 7 days. This avoided any confounding effects of the environmental conditions on the *C. giveni* larvae and enabled quantification of the loline alkaloids in the roots that the larvae consumed.

## 2. Materials and Methods

*Costelytra giveni* is a univoltine scarab insect, the larvae of which feed on roots of a range of plant species for 6 to 7 months of the year as it develops through three instars. Third instars are the most damaging stage, especially during autumn and early winter when periods of pulse drought, as represented in the experiments here, often occur.

Two trials were carried out; both used the tall fescue cultivar Jesup infected with endophyte strain AR584, and the second experiment also included meadow fescue (Northland selection breeding line) infected with its natural endophyte, strain AR1006. For both experiments individual plants were grown from germinated seed and tested for endophyte by the immunoblot method [26] when plants were at least 6 weeks old. In both experiments, plants with the appropriate endophyte status were split up to provide clones of each plant genotype across treatments to eliminate interactions between the endophyte and plant genotype. Each replicate was comprised of a different plant genotype. The endophyte status of each plant was later confirmed when the bioassays were carried out. The effects of grass host, endophyte, and interactions with environment on feeding and weight gain of third instar larvae of *C. giveni* were investigated in bioassays.

### 2.1. Experiment I

#### 2.1.1. Plant Preparation and Experimental Conditions

Tall fescue was grown from seed and tested for endophyte when 3 months old as outlined above. Ten plants without endophyte (Nil) and ten plants infected with AR584 were then split up to provide four cloned ramets of each plant, each with two tillers. Each ramet was planted into a coarsely sieved field soil in 15 cm diam. plastic pots. Plants were retained in a screenhouse for another 4 months under automatic overhead watering with regular application of fertilizer and trimming to maintain vigorous growth.

In autumn (early April), all plants were trimmed to a 5 cm residual. Of the four plants cloned for each of the 10 infected with AR584 and 10 Nil plant genotypes, two plants were transferred to a glasshouse and the remaining two were retained in the screenhouse. The screenhouse had a transparent plastic roof so that light levels were similar to the glasshouse, and mesh sides which provided the plants with ambient temperature conditions but increased exposure to wind. Each pair of plants in the two locations was either kept well-watered (WW) or drought-stressed (DS). WW plants received regular applications of measured amounts (70–150 mL) of water. The remaining plants only received water on an individual basis, when they exhibited signs of drought stress (rolled leaves and wilting). They were then given 30–50 mL of water to relieve these symptoms but not enough to make the soil moist.

After 3 weeks plants were trimmed again to 5 cm and herbage was oven dried and weighed. At the same time, soil samples were taken using a cork borer (15 mm diam. to a depth of 80 mm); soil was weighed immediately and then oven-dried at 60 °C for 48 h before reweighing to obtain an estimate of soil moisture. Throughout the experiment the temperature and humidity were monitored hourly in both locations using data loggers. Plant treatment continued until the bioassay was completed.

#### 2.1.2. Bioassay

The aim of this bioassay was to determine *C. giveni* root consumption for each replicate DS and WW plant that had been kept in either a glasshouse or a screenhouse for 26 days after the watering regimes were instigated. To do this, defined amounts of roots were provided to each of two larvae per replicate plant on two occasions over 7 days, with the remaining roots weighed after each feeding period.

Third instar larvae of C. *giveni*, were collected from a field site in Otago and transported to Ruakura Research Centre in Hamilton, where they were stored for a week at 4 °C. Healthy larvae were then selected, weighed, and two were randomly assigned for testing each plant. Whole roots (i.e., from tiller base to root tip) were removed from the base of each plant, washed and patted dry on paper towels. A 250 mg piece of whole root from each plant was placed in each of the two small plastic pots (30 mL) along with a randomly selected third instar *C. giveni* larva. Lids were placed on the pots, which were stacked randomly in replicates in a box with damp paper towels. Larvae were kept in the dark at 15 °C in a controlled environment room. After 3 days, remaining roots were removed and weighed, and another 250 mg of whole fresh root was provided to each larva. After a further 3 days, remaining root material and all larvae were weighed. The faecal pellets (frass) produced by each larva during each feeding period were dried and also weighed to validate the consumption data.

At the same time that the each of the two root samples were taken to feed larvae, additional samples were harvested from each replicate plant that were freeze dried for later loline analysis.

The data are expressed as: (i)Relative weight change (RWC) representing the change in larval weight during each 24 h period of the experiment relative to initial weight: RWC = weight gained/initial larval weight/time (days).(ii)Relative consumption (RC) of roots calculated the amount ingested for each larva over a 24 h period relative to initial weight: RC = food ingested (change in wet weight of root)/mean larval weight/number of days.

### 2.2. Experiment II

#### 2.2.1. Plant Preparation and Experimental Conditions

Both meadow fescue and tall fescue plants with and without endophyte were grown from seed in late spring in the year prior to the trial and tested for endophyte in late summer (February). Meadow fescue was infected with *E. uncinata* strain AR1006 and tall fescue with *E. coenophiala* strain AR584. Two ramets of two tillers each were split from 12 parent plants and each was planted into a 12 cm diameter pot filled with field soil in late summer. They were retained in a screenhouse under automatic overhead watering and with regular trimming and application of liquid fertiliser for 8 weeks before they were transferred to a glasshouse in mid-autumn (late April). One of each pair of cloned plants of each species was then randomly assigned to either a DS or a WW treatment. Tall fescue and meadow fescue plants were set up in separate but adjacent groups, with the WW and DS plants in randomizsed pairs. Droughted plants were not watered until they individually began to show signs of moisture stress such as wilting and rolled leaves, at which point they received 30–50 mL of water. Plants in the well-watered treatment received 80–100 mL every 2–3 days. The watering regimes continued for a period of 6 weeks until plants were harvested for the bioassay on 10 June.

#### 2.2.2. Bioassay

The bioassay was carried out in the same way as described for the previous trial. Third instar *C. giveni* larvae were collected from a field site in Canterbury, a month prior to the bioassay and stored individually in soil at 4 °C in 24 well plates. They were removed 24 h before the bioassay and two healthy active larvae were selected as replicates for each species/treatment combination (i.e., 24 larvae per treatment). Larvae were weighed and randomly assigned to treatments.

At the beginning of the bioassay, two 100 mg samples of whole root were each placed in separate 30 mL lidded plastic pots along with a weighed larva. A further two 150 mg amounts of root were also taken from each root mass and retained in sealed bags at 4 °C for use in the second feeding period. At the end of each feeding period (3 days, then 4 days), uneaten root material was removed from each pot and reweighed. Frass was also collected at the end of each feeding period and dried before weighing. Two days after the completion of the bioassay, the herbage and remaining roots were harvested from the potted plants that had continued to be WW or DS as before. Herbage and roots were freeze dried for alkaloid analysis. RC and RWC were calculated as described for Experiment I.

### 2.3. Loline Analysis

All plant material for alkaloid analysis was freeze dried soon after harvest and then finely ground. In Experiment I, equal weights of ground herbage and roots were taken from each plant and bulked across all ten replicates for each of the two samplings. Herbage and roots were harvested from Experiment II soon after the bioassay was completed. Due to the effects of the drought and the provision of roots for the bioassay there was sufficient material from droughted plants to analyze just four of the 12 replicate plants in each treatment. In addition, equal amounts of freeze-dried herbage and roots were taken from each replicate plant and combined into two bulked samples that were also analyzed for loline content.

Loline alkaloids were measured using a modification of the gas chromatographic methods reported in Baldauf et al. [27]. A sample of lyophilized grass tissue (50 mg) was extracted for 1 h with 50 µL of 40% methanol/5% ammonia and 1 mL of 1,2-dichloroethane (containing 53.7 ng mL^−1^ 4–phenylmorpholine as internal standard) followed by centrifugation at 8000× *g* for 5 min. Supernatant was transferred to a glass GC vial via a 10 µm filter for analysis. The analysis was conducted on a gas chromatography-flame ionization detector (Shimadzu GC17a; Shimadzu Corporation, Kyoto, Japan) equipped with a ZB-5 capillary column (30 m × 0.32 mm × 0.25 µm film; Phenomenex, Torrance, CA, USA). The limit of quantitation using this technique was 25 µg g^−1^.

### 2.4. Statistical Analysis

All data from the bioassays were analyzed without transformation by analysis of variance in Genstat v. 18. Soil moisture and foliar or root dry weight were analyzed using replicate, plant location (Experiment I only), and plant genotype as blocking variables. Species was used as an additional blocking variable in Experiment II as the two species were grouped separately. Plant genotype was used as a blocking variable to account for the use of cloned plants across treatments in each trial. For both trials, data on consumption, weight change, and frass output for each of the two larvae used per replicate plant were analyzed separately. Consumption and frass production for both feeding periods in each experiment were combined for analysis. Data relating to *C. giveni* that died during the bioassay (1 in Experiment I, 4 in Experiment II) were excluded from the analyses. Separate analyses were also carried out for plants in the screenhouse and glasshouse which removed the need for plant location as a blocking variable but where there were no significant differences relating to location of plants, data were pooled for analysis.

Analysis for loline concentration was only possible for Experiment II samples for four of the 12 replicate plants. Data were log transformed prior to analysis and plant genotype was used as the blocking variable for comparisons made between species and treatment.

## 3. Results

### 3.1. Experiment I

The mean temperature during the period plants were subjected to different moisture regimes was 16.2 °C in the screenhouse (Range 6.2–29.1 °C, Median 15.4 °C) and 20.6 °C in the glasshouse (Range 10.9–37.5, Median 18.9).

Three weeks after the treatments were imposed, soil moistures in DS plants were less than half of those in the WW treatment (Table 1). Despite a lower average and median temperature, soil moistures were lower in plants kept in the screenhouse than those in the glasshouse, probably because plants were also exposed to wind.

Herbage dry weight was significantly affected by endophyte, moisture, and plant location. Foliar growth of tall fescue infected with AR584 exceeded that of Nil plants under both dry and moist conditions (*p* < 0.001), and growth was higher in WW than in DS plants (Table 1). Plants kept in the glasshouse had higher herbage weights overall than those in the screenhouse, but, due to a significant interaction, differences were only significant for WW plants. No other interactions were significant.

Where trial location made no difference to the results of the bioassay, data for these parameters were pooled. In the combined analysis for plants in both locations, RC of roots from Nil plants was greater than for roots from plants infected with AR584 and greater for roots from DS than from WW plants (Figure 1a). This was reflected in significant differences in amount of root consumed in both feeding periods (data not shown) and in total, as well as for frass weight (Figure 1b). Regression analysis showed that RC accounted for 84.6 ± 5.22% of the variance in frass output (*p* < 0.001). For the screenhouse plants only, consumption of roots from DS plants was significantly reduced by endophyte whereas there was no difference for the WW plants. This endophyte by moisture status interaction occurred for RC in the first period (*p* = 0.035) and over the entire bioassay (*p* = 0.049). It also corresponded with a significant reduction in the frass output from larvae feeding on endophyte-infected DS plant roots compared with DS Nil (*df* 1/57, *F* 7.74, *p* = 0.007), whereas there was no difference between Nil and Plus WW plants. There was no indication of an endophyte x moisture status interaction for glasshouse-grown plants with endophyte significantly decreasing root consumption and frass weight under both dry and moist conditions.

On average, those larvae fed AR584 lost weight over the 6-day period of the bioassay but gained weight if fed Nil regardless of plant location, but the overall difference was not significant (Table 2). For DS plants only, however, both larval weight change and RWC were significantly lower for larvae fed roots from plants infected with AR584 compared with Nil. This difference appeared to be more pronounced for glasshouse grown plants.

### 3.2. Experiment II

Roots of both tall fescue and meadow fescue plants subjected to drought had a lower moisture content (73.2%) than well-watered (87.7%) plants (*df* 66, SED 0.77, *p* < 0.001) (Table 3). Moisture levels did not differ between species, but endophyte-infected plants had a significantly higher average moisture content compared with their Nil counterparts (Plus endophyte 81.4%, Nil 79.5%, *df* 66, SED 0.77, *p* = 0.016).

Overall, tall fescue had a greater dry root mass than meadow fescue when roots were harvested at the start of the bioassay. Root weights declined under dry conditions within each species and endophyte status combination. Root dry weights were also higher in endophyte-infected plants than endophyte-free, although for species by treatment combinations this effect was significant only for meadow fescue under moist conditions (Table 3).

RC of Nil plants was significantly higher on DS than on WW tall fescue, as occurred in Experiment I, but was not affected by moisture status in meadow fescue (Figure 1c). In interactions with species and moisture status, endophyte infection significantly reduced RC in meadow fescue irrespective of moisture status but only in DS plants of tall fescue. Similarly, RWC of larvae was also affected by interactions between endophyte, moisture, and species. *Epichloë uncinata* in meadow fescue significantly suppressed weight gain of larvae feeding on roots of both WW and DS plants, although the effect appeared stronger under WW conditions (Table 2). In contrast to this, although weight gain was reduced for larvae feeding on DS tall fescue roots from plants infected with *E. coenophiala*, this was not significantly different from the weight gain of larvae feeding on Nil DS plants. Under WW conditions the RWC was very similar for infected and non-infected plants.

Despite there being no effect of moisture status on consumption of Nil roots, frass output was significantly higher from larvae feeding on DS Nil meadow fescue and tall fescue roots than on WW roots (Figure 1d, Table 2). As for the amount of root consumed, frass production was reduced by endophyte in both plant species under dry conditions but not in tall fescue under moist conditions, a result that was consistent with Experiment I.

Dry weight of frass produced was significantly related to the fresh weight of root consumed (*p* < 0.001) but consumption only accounted for 41.2% of the variance. The relationship improved to 68.8% when the dry weight of root consumed was used as the explanatory variate. The weaker relationship between frass and consumption compared with Experiment I was likely due to the low root mass in droughted plants at the time of the bioassay. This restricted the amounts provided to each larva, resulting in a significant number of larvae consuming all roots provided. In addition, the frass produced by larvae feeding on WW roots could be expected to have had a higher moisture content than for those feeding on DS roots.

### 3.3. Loline Content

Because Experiment I samples were pooled, no statistical analysis was possible. Under screenhouse conditions, however, the DS tall fescue plants had almost three times the concentration of loline alkaloids than the WW plants (mean for two samples: DS 356 µg g^−1^, WW 128 µg g^−1^) whereas the reverse tended to happen in the glasshouse (DS 113 µg g^−1^, WW 201 µg g^−1^) although the magnitude of this difference was relatively small (Figure 2a). It was also apparent that the second sample of roots taken from each plant for the second feeding period had a lower loline content than the first samples (average of all treatments: 1st sample 259 µg g^−1^; 2nd sample 140 µg g^−1^).

Total loline concentrations, and concentrations of the loline derivatives *N*-acetylloline (NAL) and NFL in Experiment II were significantly higher in herbage from DS meadow fescue than WW plants (Figure 2b, Table 4). The opposite occurred in roots with concentration of total lolines and NANL significantly greater under WW conditions than under DS. NAL and NFL were also higher in WW roots, but the difference was not significant. These differences were also apparent in the composite samples comprising roots of all plants (mean of two composite samples: DS 201 µg g^−1^, WW 574 µg g^−1^). In tall fescue, total loline concentrations in herbage were, as for meadow fescue, significantly greater in DS than in WW herbage with this effect reflected also for NAL but not for NANL or NFL. Unlike meadow fescue, however, concentrations of total lolines and each of the loline derivatives in tall fescue roots were all higher under dry than moist conditions, although these differences were relatively small and not significant. In the composite root samples taken from all tall fescue replicate plants, however, total loline concentration in WW roots was about half that in the DS roots (84 µg g^−1^ cf. 163 µg g^−1^). NAL was largely absent in these roots but NANL was found in most samples at concentrations up to 340 µg g^−1^.

## 4. Discussion

Feeding and growth of the root-feeding scarab grub, *C. giveni*, has been affected by endophyte, moisture status, and plant species, as well as interactions between these in the two experiments reported here. We found that moisture status had contrasting effects on *C. giveni* feeding in conjunction with endophyte status. In both experiments, feeding, as measured by both RC and frass output, increased on endophyte-free tall fescue roots from DS plants compared with WW plants. Infection with *E. coenophiala* AR584 suppressed that feeding in both DS and WW treatments in Experiment I but only in DS plants in Experiment II. In meadow fescue, *C. giveni* feeding on endophyte-free plants was not affected by moisture status but was suppressed by *E. uncinata* in both DS and WW plants with a greater reduction in RC compared with Nil equivalents for WW than DS plants (% reduction for DS and WW plants respectively: RC 23% cf. 31%; frass output 17% cf. 42%). Reduced feeding due to endophyte resulted in RWC of larvae that were on average 91% less than their Nil counterparts for WW meadow fescue roots but only 40% less for DS. These results highlight strongly contrasting effects of endophyte interactions with moisture status on *C. giveni* larvae, with DS plants having the greatest effects in tall fescue and WW plants in meadow fescue.

The differences in feeding align well with the loline content of the roots *C. giveni* were feeding on. In Experiment I, loline alkaloids were markedly higher in roots of screenhouse-grown DS tall fescue plants that brought about the greatest reduction in consumption whereas, surprisingly, they were much lower in a genetically identical set of plants in the glasshouse that had no effect on *C. giveni* feeding. Irrespective of plant location, alkaloid concentrations were also lower in the second sample of roots taken to feed larvae in the later part of this bioassay and were also too low to affect the larvae (data not presented). This difference may be attributed to disturbance of the plants by removal of tillers with roots attached in order to provide larvae with equivalent root material at the start of the assay. If so, it illustrates how rapidly loline levels can change as plants respond to perturbations. The reasons for the low levels of alkaloids in glasshouse-grown tall fescue relative to the identical set of plants in the screenhouse in Experiment I are unknown. Similarly, low concentrations of lolines were found in tall fescue roots in Experiment II that was also conducted in a glasshouse at a similar time of the year. In the first experiment, there was no difference in the soil moisture status and temperature differences between the two locations were quite small (4.4 °C) so these factors are unlikely to have caused the large difference in alkaloid levels. Temperatures were not dissimilar to the conditions investigated by Kennedy and Bush [28]. These authors found a large increase in concentration of NFL and NAL in the herbage of a *Lolium/Festuca* hybrid infected with *E. coenophiala* subjected to prolonged drought and a moderate temperature regime of 21/15 °C whereas concentrations were reduced under higher and lower temperatures. It is worth noting here that loline alkaloids in total, and specifically NAL and NANL, were higher in meadow fescue herbage from DS than WW plants in our experiment, a similar result to that of Kennedy and Bush [28], but also highlighting the opposite effects found in the roots in which concentrations were higher on WW than DS plants. Comparing cloned pairs of tall fescue plants, Nagabyhru et al. [29] found increased concentrations of total lolines in shoot and roots within 2–3 days of imposition of drought. The relatively few published reports of loline levels in roots of tall fescue suggest that concentrations are typically less than 250 µg g^−1^ [2,30] although much higher concentrations of up to 700 µg g^−1^ have been recorded [1].

Meadow fescue infected with *E. uncinata* had a much stronger effect on *C. giveni* larvae than *E. coenophiala* in tall fescue in accordance with its much higher loline content. Herbage concentrations averaged 6363 µg g^−1^ in meadow fescue, 11 times the average of 570 µg g^−1^ in tall fescue. In contrast to this, there was only a four-fold difference in root concentrations between the two species (450 and 144 µg g^−1^). The opposing effects of moisture status on endophyte-mediated *C. giveni* larval feeding on tall fescue and meadow fescue were mirrored in similar effects on loline alkaloid concentration. These alkaloids were significantly elevated in roots of DS tall fescue in Experiment I (356 µg g^−1^ in DS cf. 128 µg g^−1^ in WW), and conversely in WW meadow fescue plants in Experiment II (663 µg g^−1^ in WW cf. 237 µg g^−1^ in DS). In each case, the high levels have effectively reduced root feeding and weight gain of larvae. Previous work has also shown that endophyte in meadow fescue inhibits root feeding and reduces weight gain of *C. giveni* larvae [7]. Rather surprisingly, however, the comparatively low loline concentrations in roots of DS tall fescue in Experiment II were also sufficient to reduce larval feeding in the bioassay.

Although all loline derivatives increased in WW meadow fescue roots, NANL showed the largest increase from an average of 3.5 µg g^−1^ in roots of DS plants to 72.5 µg g^−1^ in WW plants. This was surprising as a similar increase was not apparent in herbage. NANL is the most prominent constituent of the lolines produced by AR584 and was the only loline derivative that increased in roots under droughted conditions in tall fescue in Experiment II. We have no knowledge of the deterrent or possible toxic effects of NANL on *C. giveni* or what processes may regulate its concentration in roots in relation to drought. There was no indication that the low mortality that occurred in both bioassays was endophyte related. Using artificial diets, Patterson et al. [31] demonstrated that NFL and NAL significantly reduced feeding of *P. japonica* larvae at 100 µg g^−1^ with increasing inhibition at higher concentrations. Peramine, the other alkaloid produced by *E. coenophiala*, had no effect. *Costelytra giveni* larvae are known to be sensitive to loline alkaloids at concentrations ≥250 µg g^−1^ in crude extracts of tall fescue (cv. KY31) seeds [32]. In a field study with and without *C. giveni* present, Patchett et al. [33] reported very high root concentrations of between 1509 and 1935 µg g^−1^ in roots exposed to *C. giveni*, an increase of 26% compared with the same plant/endophyte associations with no *C. giveni*. The authors hypothesized that alkaloids had been reallocated from the crowns in response to larval feeding. Our study suggests that concentrations as low as 150 µg g^−1^ may be sufficient to provide a low level of feeding inhibition and >200 µg g^−1^ may reduce feeding sufficiently to affect larval growth. In bioassays carried out by Barker et al. [8], larvae of African black beetle (*Heteronychus arator*) were increasingly deterred by concentrations of lolines above 175 µg g^-1^, but that feeding did not cease even at concentrations of 5600 µg g^−1^. Further research is needed if we are to exploit this information in order to reduce the substantial damage that both insects cause to pasture in New Zealand.

Given the isolation of *C. giveni* from the direct effects of drought through the use of bioassays using excised roots here, the effect of other defense factors associated with root herbivory and endophyte infection such as increases in phenolics or induced phytohormones such as salicylic acid [34,35] can be ruled out. Nevertheless, it does not exclude the possibility that chemical changes have been initiated via the plant-endophyte association such as an increase in proline, phenolics, flavonoids, and water-soluble carbohydrates in response to abiotic stress [29]. Nguyen et al. [36] posited that crosstalk between phytohormones synergizes signaling resulting in enhanced resistance to insects, although such effects are likely to be species-specific. For plants hosting an endophyte, such complex interactions with abiotic stress may have influenced larval feeding, including the increased consumption of DS roots and potentially the production of alkaloids. Proline is one of the structural components of the biosynthetic pathway for loline production [1] but the significance of this in relation to increased drought tolerance and alkaloid production is not known. Nitrogen in leaves and roots has been shown to increase in water stressed barley plants and thereby affect herbivory [37]. A review of the plant stress hypothesis [38], however, concluded there was insufficient evidence that intermittent water stress enhanced the performance of chewing insects and where it did occur such effects may be negated by allelochemicals. For endophyte-free tall fescue in both experiments, increased feeding on roots from DS plants supported higher weight gains of larvae compared with those feeding on WW plants. On the other hand, moisture status had no influence on feeding or weight gain of larvae feeding on Nil meadow fescue plants suggesting such effects can be species specific.

Although there is considerable evidence for endophyte conferring drought tolerance in tall fescue, the vast majority of these studies have investigated the common toxic strain which, as well as loline alkaloids, produces the mammalian toxin ergovaline, the cause of fescue toxicosis in grazing livestock in the USA. Two novel strains, AR542 and AR584 (known as MaxQ™ (USA) or MaxP™ (New Zealand and Australia), that do not produce this alkaloid have been commercially released. AR542 differs in its alkaloid profile because it produces only NANL and neither NFL nor NAL, whereas AR584, used in this study, produces all three loline derivatives. *Epichloë uncinata* also produces all three alkaloids but in much higher concentrations in herbage than occurs in tall fescue. There is far less published information on the drought tolerance relating to the novel tall fescue endophytes or to *E. uncinata*. Furthermore, although *E. uncinata* is known to confer resistance to root feeding scarab grubs, there is little published information regarding AR584. A census of insects in field plots of tall fescue cv. Kentucky31 infected with the common toxic endophyte and both novel strains, found that *P. japonica* grubs weighed less, and densities of *Cyclocephala* spp. were lower in endophyte-infected plots, irrespective of strain, than in an endophyte-free control [39]. Results from our trials here also show that AR584 suppresses feeding by scarab larvae but that those effects may be enhanced under short term drought conditions that increase loline concentrations in roots. Conversely, damage to endophyte-free tall fescue may increase. Further work would be needed to prove this in the field.

Root herbivory by insects can have severe consequences for their host plants, particularly where there are resource limitations such as drought or nutrients, with estimations that the combined effects can exceed that of herbivory alone by a factor of two or more [40]. Short periods of drought are not uncommon during the summer-autumn period in temperate regions where tall fescue and meadow fescue are grown as forage grasses. In these habitats, root-feeding larvae of scarab insects are common pests capable of causing extensive damage. Our results suggest that under pulse drought conditions, increased root feeding on endophyte-free tall fescue may exacerbate damage but that effect can be reversed by the endosymbiont *E. coenophiala* producing increased concentrations of loline alkaloids. In contrast to *E. coenophiala*, meadow fescue hosting *E. uncinata* produced more alkaloids under WW than DS conditions which reduced *C. giveni* feeding to a greater extent. The different responses may reflect the environmental conditions in which these endophyte/host genotype symbioses have evolved. Further work to understand these interactions between plant species, endosymbionts, and root-feeding scarab grubs would be beneficial.

## Figures and Tables

**Figure 1 microorganisms-08-00997-f001:**
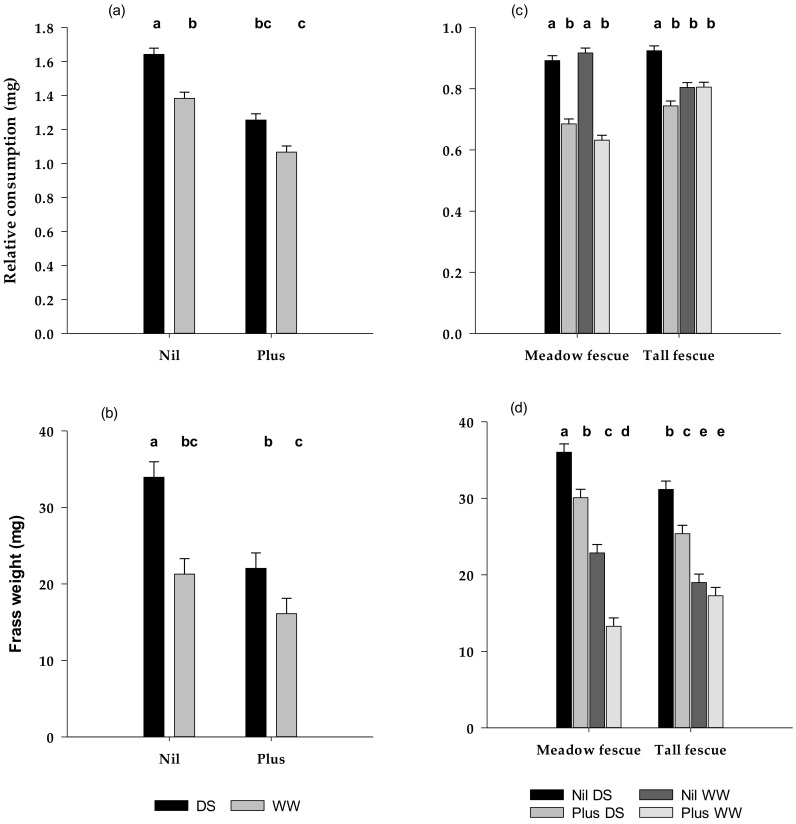
Relative consumption and frass output (mg larva^−1^) (±SE) for *C. giveni* larvae fed (**a**,**b**) roots from drought stressed (DS) and well-watered (WW) tall fescue plants with (Plus) and without (Nil) endophyte in Experiment I, and (**c**,**d**) roots from drought stressed (DS) and well-watered (WW) tall fescue and meadow fescue plants with (Plus) and without (Nil) endophyte in Experiment II. Treatments are significantly different where there is no letter in common.

**Figure 2 microorganisms-08-00997-f002:**
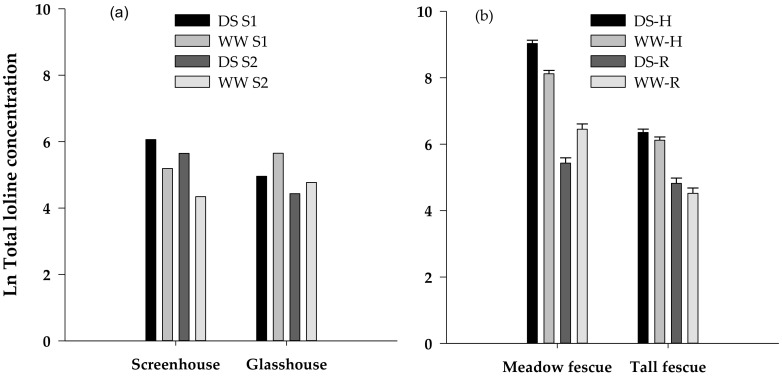
Natural log of the total loline concentration (µg g^−1^) in (**a**) two samples of roots (S1 and S2) of drought stressed (DS) and well-watered (WW) tall fescue fed to *C. giveni* over two feeding periods in Experiment I and (**b**) tall fescue and meadow fescue herbage (H) and roots (R) harvested after completion of the bioassay in Experiment II.

**Table 1 microorganisms-08-00997-t001:** Percent soil moisture and foliar dry weight of drought stressed (DS) and well-watered (WW) tall fescue with (Plus) and without (Nil) endophyte in Experiment I. Bold lettering denotes significant differences between Nil and Plus treatments.

Plant Location	DS Plus	DS Nil	WW Plus	WW Nil	*df*	*F*	*p*
% Soil Moisture							
Glasshouse	10.9	12.7	**26.6**	**27.4**	Location × moisture status		
Screenhouse	10.2	10.7	**22.3**	**21.9**	1.36	7.01	0.012
Foliar Dry Weight (g/plant)							
Glasshouse	**1.65**	**1.23**	**2.86**	**2.46**	Location × moisture status		
Screenhouse	**1.56**	**1.21**	**2.19**	**1.78**	1.36	11.21	0.002

**Table 2 microorganisms-08-00997-t002:** Comparisons of frass weight (FW) (mg larva^−1^), relative weight change (RWC), and weight change (WC) (mg larva^−1^) of *C. giveni* larvae fed roots of drought stressed (DS) or well-watered (WW) plants of tall fescue (TF) and meadow fescue (MF) in Experiment I and II. Bold lettering denotes significant differences between Nil and Plus endophyte treatments.

Expt I	DS Plus	DS Nil	WW Plus	WW Nil	Interaction	*df*	*F*	*p*
FW	**22.0**	**33.9**	**16.1**	**21.3**	Endo	1.19	14.92	0.001
					Moisture	1.117	26.5	<0.001
RWC	**−0.0017**	**0.0037**	−0.0003	−0.0003	Endo × Moisture	1.117	4.05	0.040
WC	**−2.31**	**3.76**	−0.35	−0.63		1.117	5.38	0.022
**Expt II**	**MF Plus**	**MF Nil**	**TF Plus**	**TF Nil**	**Interaction**	***df***	***F***	***p***
FW DS	**30.1**	**36.0**	**25.4**	**31.2**	Endo × Moisture × Species	1.136	7.04	0.009
FW WW	**13.3**	**22.9**	17.3	19.0				
RWC DS	**0.0093**	**0.0155**	0.0098	0.0124		1.136	5.08	0.026
RWC WW	**0.0013**	**0.0151**	0.0079	0.0074				
WC DS	**9.2**	**15.1**	9.4	11.6		1.136	5.91	0.016
WC WW	**0.8**	**14.7**	7.2	7.0				

**Table 3 microorganisms-08-00997-t003:** Percent water in roots and root dry weight of DS and WW tall fescue and meadow fescue plants with (Plus) and without (Nil) endophyte in Experiment II and significant interactions relating to moisture treatment, species, and endophyte. Bold lettering denotes significant differences between Nil and Plus treatments within the same species.

Moisture Status	MF Plus	MF Nil	TF Plus	TF Nil	*df*	*F*	*p*
% Water in Roots							
DS	**73.8v**	**70.5**	**73.8**	**74.8**	Moisture		
					1.66	352.5	<0.001
WW	**89.2**	**86.0**	**88.8**	**86.7**	Endophyte		
					1.66	6.16	0.016
Root dry weight (g/plant)							
DS	0.085	0.104	0.183	0.246	Moisture × species × endophyte		
WW	**0.386**	**0.179**	0.283	0.322	1.66	9.08	0.004

**Table 4 microorganisms-08-00997-t004:** Natural log concentrations (µg g^−1^) of *N*-acetylloline (NAL), *N*-acetylnorloline (NANL) and *N*-formylloline (NFL) in herbage and roots of drought stressed (DS) or well-watered (WW) tall fescue and meadow fescue plants. Bold lettering denotes significant differences within species due to moisture status.

Loline	Meadow Fescue		Tall Fescue		Species × Moisture Interaction			
	DS	WW	DS	WW	*df*	SED	*F*	*p*
**Herbage**								
Total	**9.029**	**8.116**	**6.353**	**6.119**	6	0.2759	5.57	0.056
NAL	**6.966**	**6.210**	**3.569**	**3.281**	6	1.1574	4.46	0.079
NANL	7.267	6.632	5.557	5.429	6	0.4477	4.77	0.072
NFL	**8.669**	**7.645**	3.982	3.802	6	1.3210	6.59	0.043
**Roots**								
Total	**5.43**	**6.45**	4.82	4.52	6	0.428	7.26	0.036
NAL	2.31	3.90	1.83	1.45	6	0.928	4.33	0.083
NANL	**0.68**	**4.24**	4.10	3.27	6	1.039	14.98	0.008
NFL	5.327	6.25	2.879	2.818	6	0.9762	6.43	0.044

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
