# Peer review of "Root Herbivory: Grass Species, Epichloë Endophytes and Moisture Status Make a Difference"

_microorganisms, 2020, doi:10.3390/microorganisms8070997_

Round 1

Reviewer 1 Report

The manuscript by Popay et al., reports a study evaluating the effects of endophyte infection, moisture status, and loline concentrations on root herbivory.   The study is well  done and the paper is well written.  It is particularly interesting that there are some differences between the E. coenophiala and E. uncinata symbioses in the effects.  A previous study (Plant Signaling & Behavior 5:1419, 2010) found that regulation of loline biosynthesis was different between E. uncinata and E. festucae. The authors may wish to include this information in the current paper. I have a few minor suggestions:

Line 157: bottle

Line 172: remove (2011)

Line 214: Should that refer to both Tables 2 and 3 since information on frass is only in Table 3

Table 2: Define DR and MO; are these DS and WW?

Line 316: plants

References: Check the references carefully for format and make them consistent.  Journal names in some are abbreviated and in some are not, some italicized and some not.  All words in some titles are capitalized and are not in some.  In several entries the E in Entomol has an accent; is that correct?

Ref. 36: Plant Mol Biol

Author Response

All suggested corrections have been made except:

L157 Change from 'pottle' to 'bottle' - the term pottle was used specifically to describe small plastic pots - the terms was also used in L154 and L120. To avoid confusion the term has been changed to pots.

L214 I have not included Table 3 here as Table 2 only includes data from Experiment I

References have been checked and changed where needed. Note that the appropriate MDPI EndNote style had been used.

Reviewer 2 Report

Review of the manuscript “Root Herbivory: Grass Species, Epichloë Endophyte and Moisture Status Make a Difference” submitted to Microorganisms, microorganisms-804742

This manuscript study discovered the complex interactions of soil moisture and Epichloë infection that affect root herbivory in tall and meadow fescue cultivars. Root herbivory from scarab larvae was reduced proportionally to increasing levels of loline alkaloids produced by the endophytes. However, soil moisture affected loline concentrations in roots differently in the two symbiota. Well-watered meadow fescue had much higher loline levels in roots than drought-stress plants, but this difference was not apparent for tall fescue plants. The study reported species specific effects of soil moisture on alkaloid distributions between shoots and roots. Moreover, root herbivory might be affected by soil moisture in uninfected plants. This study is well designed and executed. I especially appreciate that chemical analyses were performed for all individual loline compounds, which helped to explain the results from the insect bioassays. In some parts, I feel that it was hard to follow the descriptions. It sounds more like a list of facts but not like a story that is supported with facts.

Here are my detailed comments:

L14 Consider adding “also”.

L21-22: It sounds different than in L305, “Unlike meadow fescue, however, although concentrations of total lolines and each of the loline derivatives in tall fescue roots were all higher under dry than moist conditions, these differences were relatively small and not significant.”

L71: What about other differences that existed, besides temperature?

Materials and Methods: Biology of C. giveni is missing. Some information on this species will help understanding the bioassays’ design and results.

L101: It is unclear what differences existed between conditions in the greenhouse and the screenhouse? What does the screenhouse look like? What light settings were used?

L114-131: This is hard to follow. For example, it took me a while to realize that there were 2 larvae individually tested per each plant. Where was this assay performed? What were the temperature, light and humidity conditions? It will be helpful to explain why you not only measured larval weight and root consumption but also frass weight. It might be good to rewrite this section.

L119-120: “a small plastic pottle (30 mL)”  Is it better to use bottle? Pottle is ½ gallon jar which is about 1.9 liters.

L125-129: Do you think that larvae changed weight and consumed feed evenly across 6 days of the experiment? Why did you not use numbers over the experimental time period and not per day?

L129: In RC calculation for each larvae over a 24 h period, it is unclear what is the mean body mass.

L163: Was it a whole plant material or just roots or shoots?

L218 “P=0.007’, is this correct?

Figure1 and Figure 2: Mark significant differences. With this you may remove some text from the Results.

Table 2: You have term “moisture” for soil moisture treatment and for root water content. Consider using “% water in roots”.

Table 3: Confused with the right part (Interaction, df, F, P) of the table for the experiment I. Is everything correct there? Why do you have 4 comparisons for 3 variables?  Are the two upper lines not for endoXmoisture interaction?

Figure 2: For Y-axis title, should it be “Ln total loline concentration?”

Figure 2 and Table 4: Natural log concentrations of alkaloids are good for analyses but not informative when you need to compare with the other studies. Consider showing the actual alkaloid concentrations.

L350: Add measurement units after 6363.

Discussion: Consider making subchapters.

L378: Could loline alkaloids be insecticidal for C. giveni larvae? Do you think that dead larvae in you experiments might be related to loline levels? What are insecticidal concentrations of lolines for the other insect herbivores?

Author Response

Change made in Abstract (L21 & 22) to clarify what was seen as an anomaly in wording there compared with L. 305.

Methods have been rewritten in order to make them clearer and less like a 'list of facts'. Also additional information has been added as appropriate e.g. information on the biology of C. giveni, differences between the screenhouse and glasshouse.

L125-129. In Experiment I the amount consumed was very similar for both periods. In Experiment II the amount consumed was greater in the second feeding period when larvae were supplied with more roots. The standard formula for relative consumption is based on a per day amount consumed which makes it more comparable to other data that uses this same parameter and also takes account of any effect initial size of larva may have had on feeding.

L218 P=0.007 This is correct but sentence clarified as this referred to analysis of plants in the screenhouse only.

Table 3: The comparisons and interactions align with the parameters listed; i.e. for FW (Frass Weight) there are significant independent effects of endophyte and moisture, whereas for both RWC and WC there were significant endophyte x moisture interactions.

Use of subheadings in Discussion: My preference is not to use subheadings particularly for this work as there were so many interactions but to discuss the implications of the results as a whole. 

Use of actual loline concentrations rather than the log-transformed: This would be difficult to do in Fig2 as the difference in herbage concentrations between tall fescue and meadow fescue is very large. Also the data analyses as presented in Table 4 were carried out on the transformed data so would rather leave that as is and it is difficult to add in the actual concentrations without cluttering up the table. I agree that giving the reader some indications of the actual concentrations is useful and hence I have referred to these in the Discussion rather than the log-transformed ones.

Other more minor changes made:

L14 'also' added

L129 - body mass changed to larval weight

L163 - whole root

L350 - Units added

Fig 1: lettering added to indicate significant differences

Fig 2: y axis has Ln added

L378: Comment added re dead larvae

Reviewer 3 Report

This paper described the interaction between drought endophyte and grass species on insect attack. This is explored through the results of two experiments. However, as written, the comparable conclusions from the two experiments are hard to extract from the paper, and it would benefit from re-working to enable the reader to understand how the two experiments combine to produce the conclusions.

There are two experiments described. As written it is confusing to understand how each experiment is set up, and the differences between the two experiments. A flow chart of the experimental set up for plant growth, plant subdivision into ramets, the subsequent growth, water treatment, and harvesting of tissue for bioassays for the experiment with numbers of samples taken (and indications where the two experiments differ) would aid both understanding and interpretation. For example, in the current description experiment 2 describes using whole roots of 100mg with roots kept in sealed bags at 4 degrees for 3 days for the second part of the feeding. In experiment one it describes 250 mg of root being provided (whole root?)  and another 250 mg of fresh root after 3 days (was this stored at 4 degrees in plastic bags for 3 days?).

The sampling of roots for loline analysis is inconsistent between the two different experiments. In experiment 1 different roots than those used for the bioassay were taken  at the time of  harvest for loline analysis. My understanding from the descriptions  in the methods is that the remaining roots from the feeding bioassay were used.  Was this only after the second feeding? If so they had been detached from the plant for 7 days, and stored at 4 degrees for 3 days. What is the effect of this on lolines? In the section on loline analysis it states that roots were taken from the replicate plants and combined into two bulked  samples  which were also analysed. How do these compare to the bioassay samples? Which 4 four plants had sufficient samples, was it the same for tall fescue and meadow fescue?

In  the results section line 214 refers to table 2 for frass  weight, should be table 3. While the experiments are separate it would  be good to combine the  graphs in figure 1 to compare between experiments. The number of samples in each category would  also aid understanding.

Table 3 would also benefit from a consistency

in structure for comparing between samples. Ie have ds E+ and E- and ww E+ and E- tall fescue columns for experiment two, and extra columns for meadow fescue.

In figure 2 tall fescue was grown in the glasshouse in both experiments, but the root concentrations are affected in opposite directions by drought in the two experiments. How do the bioassay samples in expt 2 tall fescue compare to those from the harvested plants, which from my understanding are similar in history to those measured in experiment 1.

In the discussion it states that low levels of alkaloids in glass house plants in experiment 1 is not known line 334. Are the endophytes stable across multiple ramets over time, have there been any measurements of endophyte status in individual ramets?

In line 381 there is a statement the direct effects of drought are isolate from the larvae through the use of excised roots. However the plants had experienced ~ 26 days of drought inducement, so it is likely that the excised roots will display phenotypic adaption to the drought.

Author Response

The experiments described are two quite separate experiments. In both, endophyte in tall fescue has significantly reduced feeding with effects greater in DS then WW plants. For meadow fescue the effects were opposite. There are other interactions which are described in the results. I accept that that these could have been more clearly stated and have made changes to paragraphs L224 -237 and L267 - 274 which I think makes these results clearer.

The differences between the two experiments relate to the bioassays, rather than the way the plants were grown and treated etc. Thus I don't think this necessitates a flow chart. In the bioassays we know very little about what affects loline concentration in plants and even less about amounts in the roots so I don't think a flow chart will aid interpretation. I have made some clarifications in the text and added information to the caption for Fig 2 which I think will make the methods clearer.

Again I have changed some of the text to clarify sampling of roots for the bioassays. The number of samples = the number of replicate plants with each sample tested against two larvae - I think this is now clear in the methods. Whole roots were sampled in  both bioassays. Samples taken for loline analysis were sightly different between the bioassays as described. We did not determine effects of storage for 3 days on the loline concentration in the roots for Experiment II. Roots for loline analysis were only sampled from the potted plants at the end of the experiment (9 days after the roots were harvested for the bioassay) rather than during the experiment as in Expt I. I think this is clear in the Methods as described. We do not know how these differences in sampling affected the loline concentrations. The plants with sufficient roots for analysis at the end of the experiment were not the same replicates for tall fescue and meadow fescue.

L214 Correction made to Table 3.

The graphs are placed side by side to enable comparison. I would prefer not to combine them. Unsure what is  required re the "number of samples in each category". If this refers to the data in Figure 1, the number of samples are given in the methods: 10 replicate plants in Expt I and 12 in Expt II with two larvae tested on roots of each plant.

Regarding Table 3: I believe it would confuse the reader if the tall fescue results from two separate experiments were placed together and the meadow fescue results put in another part of the table. The value of Expt II is in the comparison between the two species.

Yes the plants in bioassays for both experiments had a similar history but experiments were undertaken at slightly different times and in different years so it is difficult to compare them directly. In addition the differences in Fig 2 for loline concentration in tall fescue were not significant so we cannot say for sure that the results for the two experiments are opposite.

Yes good point regarding the stability of endophyte infection. I had forgotten to mention that in both experiments the endophyte status of the plants was confirmed at the end of each experiment when plants were harvested. This has been added to the methods.

Phenotypic plasticity - yes this is an interesting thought and may well have occurred during the experiment II and differed between the species. However I don't think this would have affected the bioassay results as the larvae feed by severing the roots.

Round 2

Reviewer 3 Report

My comments to the previous version were aimed at enabling the reader to compare the two different experiments, done at different times to understand what is consistent between the two experiments.

If the effect of drought on soil moisture, plant growth, plant organ weight, plant organ dry weight and loline levels is consistent in tall fescue across two experiments, then understanding how the endophyte status and feeding interact with this drought stress is clearly shown, and more inference can be placed on the meadow fescue experiment. If the drought and its effects on plant growth are consistent, but the growth in a shade house verses glass house verses glass house later in the month in a different year show consistent drought effects but different feeding effects, are the right variables being measured and controlled?

Comparing the two experiments with similar measurements for both will enable the reader to understand whether there are some measured variables that are comparable between the two experiments, that may explain where there are some different or non-significant results in the data from tall fescue that do not agree between the two different experiments on this species. The repeatability of the tall fescue results is important in understanding the weight to be given to the single experiment conducted with meadow fescue.

As tall fescue plants were in effect treated very similarly in the two experiments, one would expect similar trends in the two experiments.

For experiment 1 in table 1 % soil moisture and Foliar dry weight per plant are presented. Are such values not available for experiment 2?

For experiment 2 we have root dry weight, and percent water measurements, are these values not available for experiment 1?

For the loline measurements, in experiment one for tall fescue, in the screenhouse lolines are higher in drought stress in the screenhouse compared to well-watered, but lower in drought stress in the glasshouse compared to well-watered. This lower level of lolines in drought stressed roots compared to a well-watered treatment in the glasshouse is the same trend as seen for meadow fescue roots in experiment 2.

Line 235-237 states “There was no indication of an endophyte x moisture status interaction for glasshouse-grown plants with endophyte significantly decreasing root consumption and frass weight under both dry and moist conditions.” The experiment described as the second experiment is undertaken in a glasshouse.

Data is easier to compare in graphs, annotated with statistical significance where there is some.

If frass weight is shown as a bar graph for both experiments it is visualised that this is always lower in E+ plants in both experiments, and where significant, it could be marked with an Asterix.

Author Response

Changes have been made to address the concerns of the reviewer. I have added Frass output to Figure 1 so that this can be directly compared to consumption. I have also added lettering so that significant differences are clear. 

In the text I have added comments to show where there is consistency in the results. e.g. for tall fescue relative consumption in both experiments was higher on DS than WW Nil plants. In both experiments C. giveni feeding was reduced on DS roots and not on WW roots (significant for pooled data and screenhouse plants). The results were the same for frass output and supported by the  weight change of larvae. 

There was consistency also in the effects of endophyte on consumption and frass output in meadow fescue in the second experiment. 

These effects are consistent with differences in loline alkaloid concentration, with DS tall fescue having higher loline alkaloids than WW tall fescue although there are some anomalies in the data e.g. glasshouse data in Expt I that cannot be explained (note that these differences were quite small). Similarly, the higher loline concentration found in WW meadow fescue compared with DS meadow fescue is consistent with effects on larvae. In addition data on the limited number of replicates in Expt II that could be analysed were supported by the data from two composite samples from all plants in the same treatments that were analysed. It should be noted that there is a high degree of variability in alkaloid production from individual plants which cannot all be accounted for by plant genotype; hence a lack of consistency is not unusual.

In response to other comments: 

Re percent soil moisture: The pots used In Experiment II were smaller than those in Experiment I which made soil sampling problematic. The root sampling and % water in the roots provided information to illustrate that the plants were drought stressed as did the % soil moisture and differences in foliar dry weight in Experiment I.

Line 235-237: There was a significant effect of endophyte in the glasshouse grown tall fescue in Experiment I regardless of moisture status and a significant effect in Experiment II only for DS plants. These were experiments conducted in different years and although at a similar time of the year, the effects on loline production are likely to vary because of other factors. Unfortunately not all effects can be accounted for.

Round 3

Reviewer 3 Report

The changes to the paper have now made it much easier to understand how the two separate experiments compare to each other, and the basis for drawing the conclusions stated in the paper.